

# Predicted distribution of a rare and understudied forest carnivore: Humboldt marten (*Martes caurina humboldtensis*)

Katie M. Moriarty[1,*], Joel Thompson[2,*], Matthew Delheimer[3], Brent R. Barry[4], Mark Linnell[5], Taal Levi[6], Keith Hamm[7], Desiree Early[7], Holly Gamblin[8], Micaela Szykman Gunther[8], Jordan Ellison[1], Janet S. Prevéy[9], Jennifer Hartman[10] and Raymond Davis[11]

[1] Western Sustainable Forestry, National Council for Air and Stream Improvement, Inc., Corvallis, OR, United States of America
[2] Pacific Northwest Region Data Resources Management, USDA Forest Service, Joseph, OR, United States of America
[3] Pacific Southwest Research Station, USDA Forest Service, Placerville, CA, United States of America
[4] Confederated Tribes of the Grand Ronde, Grand Ronde, OR, United States of America
[5] Pacific Northwest Research Station, USDA Forest Service, Corvallis, OR, United States of America
[6] Department of Fisheries and Wildlife, Oregon State University, Corvallis, OR, United States of America
[7] Green Diamond Resource Company, Korbel, CA, United States of America
[8] Department of Wildlife, Humboldt State University, Arcata, CA, United States of America
[9] Fort Collins Science Center, US Geological Survey, Fort Collins, CO, United States of America
[10] Rogue Detection Teams, Rice, WA, United States of America
[11] Pacific Northwest Region, USDA Forest Service, Corvallis, OR, United States of America
[*] These authors contributed equally to this work.

Corresponding author
Katie M. Moriarty,
kmoriarty@ncasi.org

## ABSTRACT

**Background**. Many mammalian species have experienced range contractions. Following a reduction in distribution that has resulted in apparently small and disjunct populations, the Humboldt marten (*Martes caurina humboldtensis*) was recently designated as federally Threatened and state Endangered. This subspecies of Pacific marten occurring in coastal Oregon and northern California, also known as coastal martens, appear unlike martens that occur in snow-associated regions in that vegetation associations appear to differ widely between Humboldt marten populations. We expected current distributions represent realized niches, but estimating factors associated with long-term occurrence was challenging for this rare and little-known species. Here, we assessed the predicted contemporary distribution of Humboldt martens and interpret our findings as hypotheses correlated with the subspecies' niche to inform strategic conservation actions.

**Methods**. We modeled Humboldt marten distribution using a maximum entropy (Maxent) approach. We spatially-thinned 10,229 marten locations collected from 1996–2020 by applying a minimum distance of 500-m between locations, resulting in 384 locations used to assess correlations of marten occurrence with biotic and abiotic variables. We independently optimized the spatial scale of each variable and focused development of model variables on biotic associations (e.g., hypothesized relationships with forest conditions), given that abiotic factors such as precipitation are largely static and not alterable within a management context.

**Results**. Humboldt marten locations were positively associated with increased shrub cover (salal (*Gautheria shallon*)), mast producing trees (e.g., tanoak, *Notholithocarpus densiflorus*), increased pine (*Pinus sp.*) proportion of total basal area, annual precipitation at home-range spatial scales, low and high amounts of canopy cover and slope, and cooler August temperatures. Unlike other recent literature, we found little evidence that Humboldt martens were associated with old-growth structural indices. This case study provides an example of how limited information on rare or lesser-known species can lead to differing interpretations, emphasizing the need for study-level replication in ecology. Humboldt marten conservation would benefit from continued survey effort to clarify range extent, population sizes, and fine-scale habitat use.

# INTRODUCTION

Modeling predicted distributions is important to direct conservation efforts yet creating accurate predictions is challenging for rare, declining, or understudied species (*Raphael & Molina, 2007*). For instance, constriction of the range available to a species—it's realized niche—is the actualization of used conditions, but such conditions may change (*Colwell & Rangel, 2009*). Contemporary location information may further associate a species with conditions that were unaffected by prior agents of population decline, but not with favored characteristics where the species resided prior (*Caughley, 1994*). For instance, bison (*Bison bison*) were historically widely distributed throughout the Great Plains of North America (*Shaw, 1995*), yet a contemporary species distribution model would associate bison occurrence with conditions where the few relict populations reside, including the extremely cold winters and thermal geysers of Yellowstone National Park. Appropriate interpretation of the conditions that constitute suitable habitat is requisite for species' management and spatial models may help predict occurrence (*Sofaer et al., 2019*).

Humboldt martens (*Martes caurina humboldtensis*) are a distinct subspecies of the Pacific marten (*M. caurina*) that historically occurred throughout coastal forests of northern California and Oregon (*Schwartz et al., 2020*). Humboldt martens were thought to be increasingly rare almost a century ago (*Grinnell & Dixon, 1926*) and were considered to be extirpated in California and extremely rare in Oregon for the latter half of the 20th century (*Zielinski et al., 2001*). In 1996, the Humboldt marten was rediscovered in California (*Zielinski & Golightly, 1996*). Subsequent research efforts over the last two decades have elucidated some aspects of Humboldt marten ecology and demography (e.g., *Linnell et al., 2018*; *Delheimer et al., 2021*), including surveys to evaluate Humboldt marten distribution (e.g., *Gamblin, 2019*; *Moriarty et al., 2019*). Such investigations have improved our knowledge of where Humboldt martens occur yet the full geographic extent of the contemporary distribution remains unknown, although it appears to compose a fraction of the historical distribution (*USFWS, 2020*). This putative range contraction

has resulted in apparently small and disjunct populations (*USFWS, 2019*), which has engendered substantial concern for the species' persistence. Consequently, Humboldt martens were listed as Endangered under the state of California's Endangered Species Act (*CDFW, 2019*) and as Threatened under the federal Endangered Species Act as a "coastal distinct population segment" of Pacific martens (*USFWS, 2020*).

Clarifying the contemporary Humboldt marten distribution by identifying areas where martens may occur that have not been surveyed and predicting the future distribution (e.g., identifying areas where martens may not currently occur but could colonize) is urgently needed for conservation planning. Nonetheless, modeling the distribution of Humboldt martens is constrained by apparent non-stationary associations between extant populations, and vegetation associations that contradict the prevailing paradigm for North American martens. For instance, it has generally been recognized that North American martens occur in mature forests characterized by dense canopy cover, presence of large diameter and decadent trees and snags, and abundant coarse woody debris (*Thompson et al., 2012*). Although initial investigations primarily associated Humboldt martens with similar conditions (*Slauson, Zielinski & Hayes, 2007*), subsequent studies have indicated that Humboldt martens also occur in young forests (<80 years old) with modest canopy cover and relatively small diameter trees (*Eriksson et al., 2019*; *Moriarty et al., 2019*). Dense and spatially-extensive shrubs, also an uncharacteristic vegetation association for martens elsewhere in North America, was a consistent habitat component in most studies of Humboldt martens (*Slauson, Zielinski & Hayes, 2007*; *Eriksson et al., 2019*; *Gamblin, 2019*; *Moriarty et al., 2019*). Similarly, European pine martens (*Martes martes*) have long been considered a habitat specialist associated with older forests (*Storch, Lindstrom & De Jounge, 1990*; *Brainerd & Rolstad, 2002*), yet have recently been documented in a wide variety of habitat types including shrublands, grasslands, and agricultural areas (*Lombardini et al., 2015*; *Balestrieri et al., 2016*; *Moll et al., 2016*; *Manzo et al., 2018*).

Observations that are limited in space or time may not identify the conditions necessary for population persistence, which could result in a misrepresentation of a species' niche. A previous range-wide Humboldt marten distribution model by *Slauson et al. (2019)* emphasized a strong correlation between Humboldt marten occurrence and an "old-growth structural index" (OGSI) variable, which is a composite index of factors considered common to old-growth forests in the region, including density of large live trees, snags, and downed wood, stand age, and diversity of tree sizes (*Davis et al., 2015*). However, more recent and broader-scale research efforts suggest that associations between OGSI and Humboldt marten distribution are much less clear (e.g., *Barry, 2018*; *Gamblin, 2019*; *Linnell et al., 2018*; *Moriarty et al., 2019*). A potential mismatch in previously-predicted associations between vegetation and Humboldt marten distribution could lead to a "wicked problem" by focusing management or restoration in areas that may not benefit the species across its range (*Gutiérrez, 2020*).

Here, our objective was to create a contemporary range-wide model of predicted Humboldt marten distribution that includes recent location data collected from broad-scale randomized surveys throughout the historic range, combined with more recent and accurate vegetation layers (e.g., shrub layers). Our goal was to predict factors contributing

to Humboldt marten distribution and to highlight areas for future surveys and conservation efforts.

## MATERIALS & METHODS

### Study area

We collected data throughout coastal northern California and Oregon. The Humboldt marten is considered to occur in four Extant Population Areas (EPAs), which were created using minimum convex polygons around clusters of marten detections, but excluded clusters with smaller numbers of detections (<5) or detections >5 km from other detections (*USFWS, 2019*). As such, our surveys included both the recognized EPAs (Central Coastal Oregon, Southern Coastal Oregon, California-Oregon Border, and Northern Coastal California; Fig. 1) but also extended between these designated boundaries to include the historic range (*USFWS, 2019*).

Surveys in California occurred in both near-coastal and montane areas (Klamath Mountains, California Coast Range) that received substantial precipitation (100–300 cm annual precipitation) with cooler (7–10 °C) temperatures and drier summers dominated with fog and low cloud moisture (*Rastogi et al., 2016*). Forest types included a mix of coniferous and hardwood with a spatially-extensive shrub understory and dominant tree species included redwood (*Sequoia sempervirens*) along the coast and Douglas-fir (*Pseudotsuga menziesii*) in the mountains (*Whittaker, 1960*).

Surveys in Oregon similarly occurred in both near-coastal and montane areas (Oregon Coast Range) where dominant forest types included Sitka spruce (*Picea sitchensis*) and shore pine (*Pinus contorta*) along the coast and western hemlock (*Tsuga heterophylla*) slightly inland (*Franklin & Dyrness, 1973*). The Sitka spruce zone was characterized by a wet and moderately warm maritime climate with average annual temperatures of 10–11 °C, average annual precipitation of 200–300 cm, and frequent fog and cloud cover. The western hemlock zone, which was often co-dominated by Douglas-fir, was somewhat cooler (7–10 °C average annual temperature) and drier (150–300 cm annual precipitation) with fairly extensive summer fog and low cloud cover (*Dye et al., 2020*).

Common conifer species intermixed and included western hemlock, Port Orford cedar (*Chamaecyparis lawsoniana*), and western redcedar (*Thuja plicata*). Hardwood trees included tanoak (*Notholithocarpus densiflora*), giant chinquapin (*Castanopsis chrysophylla*), coastal live oak (*Quercus agrifolia*), canyon live oak (*Q. chrysolepis*), California bay (*Umbellularia californica*), red alder (*Alnus rubra*), bigleaf maple (*Acer macrophyllum*), and Pacific madrone (*Arbutus menziesii*). Dominant shrubs throughout the study area included salal *(Gautheria shallon)*, evergreen huckleberry (*Vaccinium ovatum*), Pacific rhododendron *(Rhododendron macrophyllum)*, and red huckleberry (*V. parvifolium*).

### Marten locations

We used spatially-referenced Humboldt marten locations collected between 1996 and 2020 in California and Oregon. We excluded locations occurring in areas that were modified by fire or timber harvest after the location date and prior to 2016, the date represented by our vegetation data. If multiple locations occurred within a 500-m × 500-m grid cell,

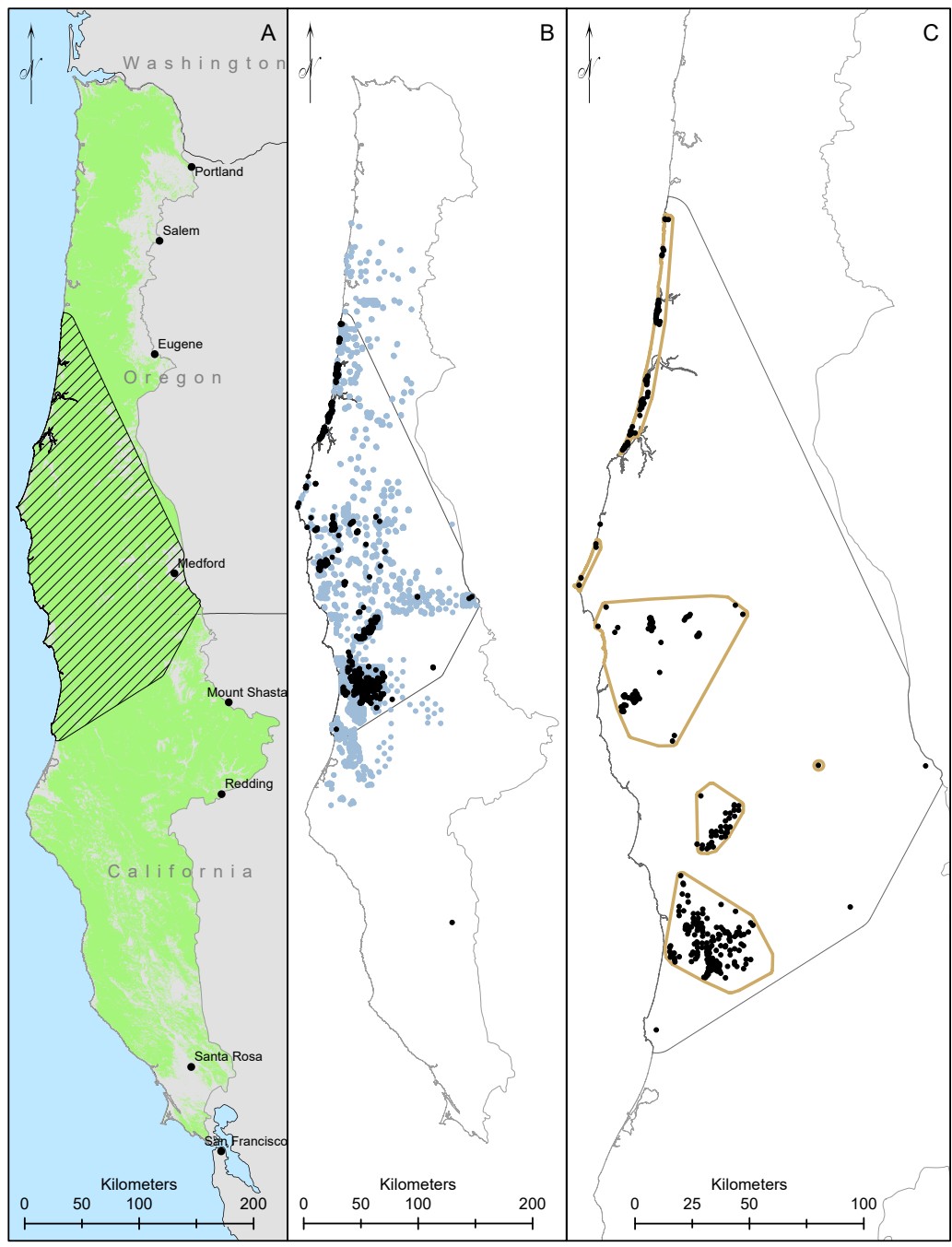

**Figure 1** **Our study area and modelling region for Humboldt martens (*Martes caurina humboldtensis*) included all of coastal Oregon and northern California.** We modeled Humboldt marten predicted distributions in forested lands ((A), green mask) in two ecoregions. We created a minimum convex polygon of known locations buffered by 10-km (hatched area). We compiled 10,229 marten locations, displaying 1,692 marten locations that were not GPS derived and clustered (icon color) from 5,153 surveyed sites with non-detections in light gray, collected during 1996–2020 (B). We spatially thinned locations to approximately 500m apart, prioritizing den and rest locations and resulting in 384 locations (black dots, (C)).

we spatially-thinned locations to randomly include one in each cell, attempting to achieve spatial independence for modeling (*Kramer-Schadt et al., 2013*). Priority for location retention from highest to lowest was: (1) rest and den locations from telemetry (*Linnell et al., 2018*; *Delheimer et al., 2021*); (2) locations from scat dog detection surveys (*Moriarty et al., 2018*; *Moriarty et al., 2019*); and (3) locations from baited camera and/or track plate surveys (*Slauson, Baldwin & Zielinski, 2012*; *Barry, 2018*; *Gamblin, 2019*; *Moriarty et al., 2019*). We used presence-only data because surveys that occurred prior to 2014 were often missing detection histories from non-detection (e.g., absence) locations.

For the data for which the authors were responsible, our protocols were reviewed and approved by the USDA Forest Service Research and Development Institutional Care and Use Committee (permits 2015-002, 2017-005) or Humboldt State University Institutional Care and Use Committee (permit 16/17.W.05-A). We obtained Scientific Take Permits for hair snares and samples collected through the Oregon Department of Fish and Wildlife (ODFW 119-15, 128-16, 033-16, 109-19, 107-20). Older verified survey data were provided by the US Fish and Wildlife Service with no additional information.

## Modeling approach

Our modeling approach included Humboldt marten locations, biotic and abiotic predictor variables, and randomly generated pseudo-absence points. We used a minimum convex polygon (MCP) around Humboldt marten locations buffered by 10 km to define the modeling region (Fig. 1B). We chose a 10 km buffer because it approximated the upper quartile of daily marten movement (*Moriarty et al., 2017*). We projected our model to available vegetation data from Gradient Nearest Neighbor (GNN) data supplied by the Landscape Ecology, Modeling, Mapping and Analysis lab (*Bell, Gregory & Davis, 2020*; *Bell et al., 2021*), which included the coastal and Klamath level-3 eco-provinces (*U.S. Environmental Protection Agency, 2013*). We removed urban areas and water from the background data (*Davis et al., 2016*). We summarized the range, average, and standard deviation for each variable within the modeling region and study area (Table 1, Fig. 1).

## Biotic variables

Biotic variables in our models included forest structure and composition, forest age, canopy cover, OGSI, percent pine, percent mast, and predicted shrub cover, as described below.

We used the 2016 version of GNN (*Ohmann & Gregory, 2002*) to incorporate forest structure variables including forest age, canopy percent cover, OGSI, and percent pine. Forest age was the basal area-weighted age based on field-recorded or modeled ages of dominant and codominant trees. Canopy percent cover was calculated using the Forest Vegetation Simulator (*Crookston & Stage, 1999*). Our OGSI index ranged from 0–100 was based from 4 elements: density of large diameter live trees per hectare, density of large diameter snags per hectare, percentage of downed wood greater than 25 cm in diameter, and an index of tree diameter diversity computed from tree densities in different diameter classes (*Davis et al., 2015*). For live trees and snags, "large diameter" was dependent on forest type and was defined for twelve vegetative zones, each zone with a unique minimum diameter threshold (i.e., ranging 50–100 cm for live trees, 50–75 cm for snags; *Davis et*

Moriarty et al. (2021), *PeerJ*, DOI 10.7717/peerj.11670

**Table 1** **Data ranges, means, and standard deviations for the model region, the contemporary Humboldt marten distribution, and at Humboldt marten locations.**
We depict individual layer statistics within our Humboldt marten (*Martes caurina humboldtensis*) model region in coastal Oregon and northern California. We display the variable, optimized spatial scale with a radius in meters, value range from the coastal ecoregions, means and standard deviation (SD) for the model region, minimum convex polygon around all known marten locations (MCP), and values from spatially thinned marten locations ($n = 384$), our layer source, and a description of that variable. We only considered variables with <60% correlation in our final model (Table S2).

| Variable | Scale | Value range | Model region (Mean ± SD) | Minimum convex polygon (Mean ± SD) | Marten locations (Mean ± SD) | Source | Description |
|---|---|---|---|---|---|---|---|
| Forest age, years | 270 | 0–712 | 95.5 ± 43 | 104.3 ± 49.4 | 109.8 ± 69.6 | 2016 GNN | Basal area weighted stand age based on field recorded or modeled ages of dominant/codominant trees |
| Canopy cover (%) | 1170 | 2–99 | 65.9 ± 13 | 66.4 ± 14 | 71.3 ± 18.6 | 2016 GNN | Canopy cover percentage of all live trees |
| Coastal proximity | 50 | 2–700 | 511.7 ± 193.1 | 516.3 ± 203.1 | 361.8 ± 197.9 | PRISM | Optimal path length from the coastline accounting for terrain blockage (*Daly et al., 2008*) |
| Diameter diversity index | 1170 | 26–811 | 433.9 ± 103 | 437.6 ± 111.7 | 459.4 ± 123.6 | 2016 GNN | Diameter diversity index - measure of stand structure based on tree densities in diff. DBH classes (x100) |
| Percent downed wood | 270 | 0–797 | 69.3 ± 54.7 | 70.9 ± 50 | 68.5 ± 60.1 | 2016 GNN (created) | Created within GNN to estimated percentage of large downed wood, a component of OGSI |
| Salal | 1170 | 0–100 | 35.7 ± 30.9 | 50.7 ± 32.3 | 72.7 ± 17.8 | Prevéy | Probability of *Gautheria shallon* species occurrence (*Prevéy et al., 2020*) |
| Masting vegetation | 1170 | 0–72 | 5.9 ± 7.4 | 5.2 ± 6.7 | 9.3 ± 9 | 2016 GNN | Percent of stand basal comprised of tanoak (*Notholithocarpus densiflorus*; LIDE), giant chinquapin (*Castanopsis chrysophylla*; CHCH), coastal live oak (*Quercus agrifolia*; QUAG), canyon live oak (*Quercus chrysolepis*; QUCH), and California bay (*Umbellularia californica*; UMCA) (mast producing evergreen hardwoods, indicator of prey abundance) |

**Table 1** (*continued*)

| Variable | Scale | Value range | Model region (Mean ± SD) | Minimum convex polygon (Mean ± SD) | Marten locations (Mean ± SD) | Source | Description |
|---|---|---|---|---|---|---|---|
| Old growth structural index | 50 | 0–100 | 32.7 ± 15.8 | 33.2 ± 16.1 | 33.8 ± 16.9 | 2016 GNN | Old-growth structure index based on abundance of large live trees, snags, down wood, and Diameter Diversity Index (DDI) |
| Percent pine | 1170 | 0–94 | 1.2 ± 3.5 | 1.5 ± 4.5 | 10.9 ± 20.1 | 2016 GNN | Percent of pixel basal area comprised of shore pine (*Pinus contorta*; PICO), Jefferey pine (*Pinus jeffreyi*; PIJE) and knobcone pine (*Pinus attenuata*; PIAT). We use this as an indicator of serpentine and coastal dune environments. |
| Percent slope | 1170 | 0–74 | 33.8 ± 10.9 | 36.2 ± 10.6 | 31.7 ± 15.8 | USGS DEM | Percent slope in degrees |
| Precipitation | 1170 | 13–198 | 66.9 ± 27 | 70 ± 30.1 | 102.4 ± 30.5 | 2016 GNN | Average annual precipitation 1981–2010 (inches) |
| Large snag density | 742 | 0–48 | 4.9 ± 4.3 | 5.8 ± 4.6 | 6.9 ± 4.9 | 2016 GNN (created) | Created within GNN to estimated density of large snags, a component of OGSI |
| Temperature (August max) | 1170 | 8–24 | 16.5 ± 2.3 | 16.1 ± 1.7 | 16.4 ± 1.7 | PRISM | Average annual maximum temperature 1981–2010 (Celcius). |
| Topographic position index | 270 | -149–174 | 0.7 ± 26.7 | 1.1 ± 28.8 | −0.3 ± 28.6 | USGS DEM | Topographic position index - difference of cell elevation with mean of all cells w/in 450 m radius |
| Large tree density | 1170 | 0–47 | 3.2 ± 3.5 | 4.4 ± 4.2 | 5.2 ± 5.9 | 2016 GNN (created) | Created within GNN to estimated density of large trees, a component of OGSI |
| Huckleberry | 1170 | 2–99 | 32.7 ± 24.6 | 39.1 ± 26 | 42.7 ± 27.2 | Prevéy | Probability of species occurrence for *Vaccinium ovatum* (created) |

*al., 2015*); see Item S1 for more information on integration of the OGSI variable into our model.

We created a variable called "percent pine", which was the combined percentage of total basal area of shore pine, Jeffreyi pine (*P. jeffreyi*), and knobcone pine (*P. attenuata*) from GNN. This variable was included because martens have been detected in shore pine communities in the Oregon Central Coast population (*Linnell et al., 2018*; *Eriksson et al., 2019*), and in areas with serpentine soils characterized by sparse cover of Jeffreyi and knobcone pine, stunted tree growth, and dense shrub understories (*Kruckeberg, 1986*; *Safford, Viers & Harrison, 2005*; *Harrison et al., 2006*; *Slauson et al., 2019*). We visually inspected the congruence of the serpentine soil layer created by the US Fish and Wildlife Service (*Schrott & Shinn, 2020*) with our percent pine layer, confirming overlap between the two variables.

Humboldt martens have been associated with dense shrub cover throughout their range (*Slauson, Zielinski & Hayes, 2007*; *Moriarty et al., 2019*). Salal and evergreen huckleberry appear particularly important, as the berries of each occur in Humboldt marten diets and provide food for marten prey species (*Eriksson et al., 2019*; *Manlick et al., 2019*; *Moriarty et al., 2019*). We modeled probabilities of species occurrence of salal and evergreen huckleberry, creating the model for evergreen huckleberry following methods published for salal and other shrub species (*Prevéy, Parker & Harrington, 2020*; *Prevéy et al., 2020*). We related locations to contemporary (1981–2010) bioclimatic variables from the AdaptWest project (*Wang et al., 2016*) to depict the probability of species occurrence (1–100%). Humboldt marten diet is dominated by animals (e.g., passerines, ground squirrels) that feed on berries and mast and Humboldt martens also directly consume berries (*Slauson & Zielinski, 2017*; *Eriksson et al., 2019*; *Manlick et al., 2019*). The "mast" variable represented hardwood tree and shrub species that produce nuts, seeds, buds, or fruits eaten by wildlife and was estimated using the 2016 GNN layer as the percent of total basal area comprised of tanoak, giant chinquapin, coastal live oak, canyon live oak, and California bay.

## Abiotic variables

Abiotic variables included temperature (°C), precipitation (cm), cloud cover (%), coastal proximity, percent slope, and topographic position index. We used 30-year normal PRISM variables of Average Annual Precipitation converted to cm and Maximum Temperature in August at an 800-m scale (1981-2010, PRISM Climate Group, Oregon State University, http://prism.oregonstate.edu, created 10/17/2019) as a proxy for maximum annual temperature. We explored annual data for temperature (2010–2018), but the available 4 km resolution produced artifacts in the model.

We created models with the variable Coastal Proximity, which uses PRISM data and combines coastal proximity and temperature advection influenced by terrain (*Daly, Helmer & Quiñones, 2003*) modified for the western United States (*Daly et al., 2008*). We derived percent slope and topographic position index from US Geological Survey digital elevation models. Topographic position index is an indicator of slope position and landform category; it is the difference between the elevation at a single cell and the average elevation of the user-defined radius around that cell (*Jenness, 2006*).

## Scale optimization

Given that martens select habitat at multiple scales (e.g., broad-scale landscape features) and fine-scale features within home ranges (4th order selection; e.g., *Minta, Kareiva & Curlee, 1999*), we optimized the spatial scale of each variable included in the model. We smoothed variables using the extract function in package *raster* in R (*Hijmans, 2020*; *R Core Team, 2020*) with a radius of 50 m, 270 m, 742 m, and 1,170 m. Our smallest scale (50 m, 0.81 ha) provided local and fine-scale conditions. We assumed 270 m (20 ha) approximated the size of a Humboldt marten core area, similar to optimized scales of vegetation characteristics used in predicting conditions for marten rest structures elsewhere in California (*Tweedy et al., 2019*). The scale of 742 m (174 ha) represented an approximate female Humboldt marten home range size, calculated as the average of female home range estimates (173 ha) from two previous studies (*Linnell et al., 2018*; Data S1; *PSW, 2019*). Our broadest scale was based on the largest size of a Humboldt marten male home range (1,170 m, 428 ha, Data S1), assuming a male would overlap multiple females and could be interpreted as the smallest unit of population level selection (*Linnell et al., 2018*; *PSW, 2019*). We used individual univariate linear models (glm) for each spatial scale using our training location data and a random background sample of 9,600 points (25 times the location data) within the MCP at different locations than the Maxent generated pseudo-absence data (Data S2). Similar to prior examples (*Wasserman et al., 2010*; *McGarigal et al., 2016*; *Zeller et al., 2017*), we selected the scale for each variable that had the most extreme, and thus the most predictive, coefficient as well as the lowest Akaike's Information Criterion (AIC) value. We also visually inspected the fit of each spatial scale using boxplots (Figs. S1–S3).

We provided boxplots to visually estimate whether our final variables were similar between all marten locations, thinned marten locations, available surveyed locations without detections (non-detection), and random locations (Fig. 2).

## Predicted distribution

We used Maxent modeling software v3.4.1 (*Phillips, Anderson & Schapire, 2006*) to estimate the relative probability of Humboldt marten presence (*Merow, Smith & Silander, 2013*). Maxent uses a machine learning process to develop algorithms that relate environmental conditions at documented species' presence locations to that of the surrounding background environment in which they occurred (*Phillips & Dudík, 2008*; *Elith et al., 2011*). We excluded variables with highly correlated predictors (|Pearson coefficient|>0.6), selecting the variable that was most interpretable for managers (Table S2). During this process, we considered the variance inflation (Table S3), which allows for evaluation of correlation and multicollinearity. Variance inflation factors equal to 1 are not correlated and factors greater than 5 are highly correlated as determined by $(1/(1-R_i^2))$, where $R_i^2$ is squared multiple correlation of the variable *i* (*Velleman & Welsch, 1981*).

Within each model iteration, we selected the bootstrap option with 10 replicates, random seed, and 500 iterations. We trained our models using a random subset of 75% of presence locations and tested these using the remaining 25% with logistic output. We used the default of 10,000 pseudo-absence background samples. We varied the response functions to include linear, product, and quadratic features. We selected the "auto features" option

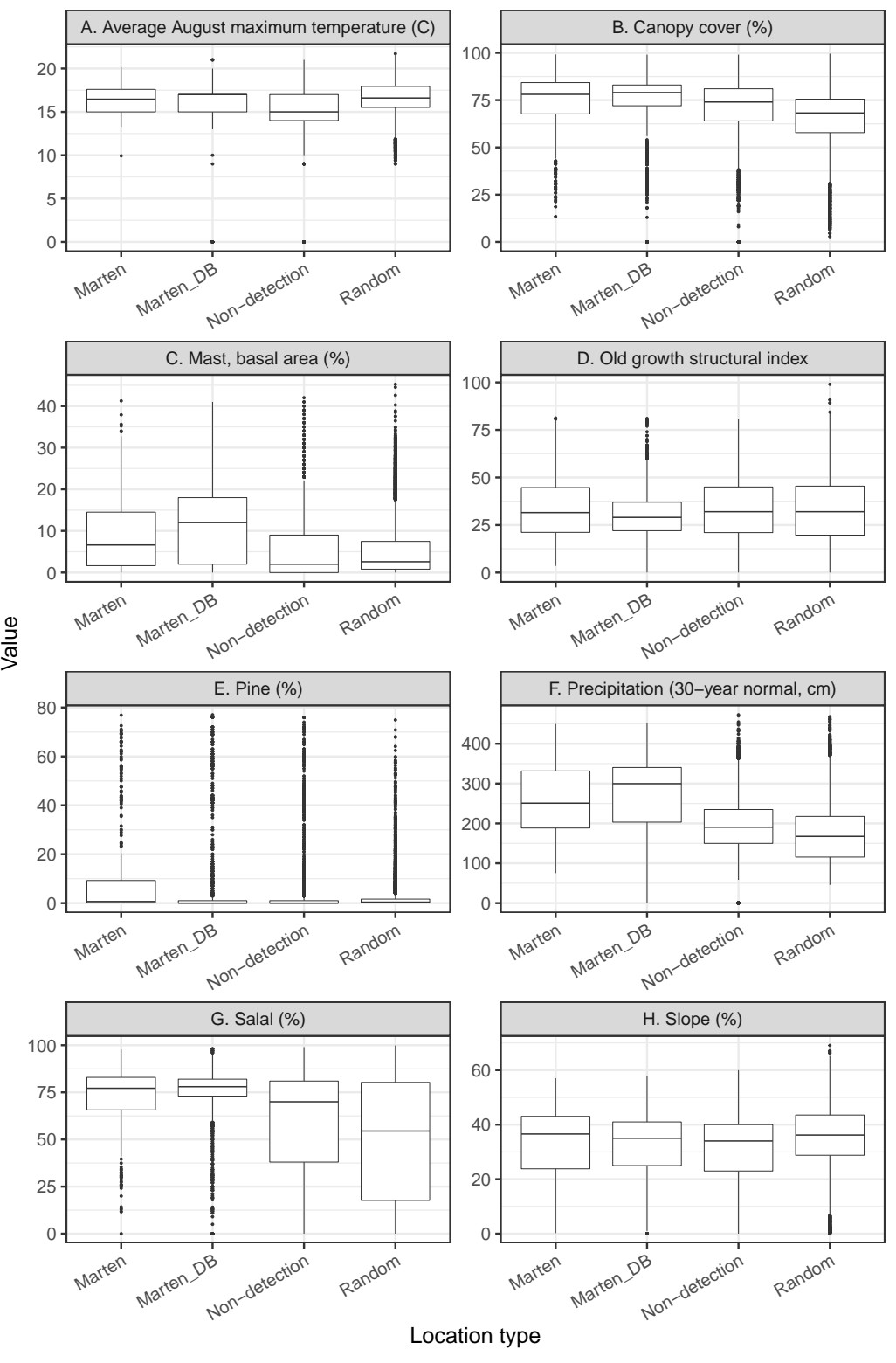

**Figure 2** **We investigate the range of variables in our thinned dataset compared to all marten locations and detection/non-detection data (A-H).** To provide the range of values observed in this study, we depict boxplots for the variables in the top model showing the thinned marten data (Marten), all non-GPS marten locations (Marten_DB), non-detected but surveyed locations (non-detection), and random locations within the minimum convex polygon (9,600 random locations).

for all runs, which allows Maxent to further limit the subset of response features from those selected by retaining only those with some effect.

Species distribution maps were produced from all models using the maximum training sensitivity plus specificity threshold, which minimizes both false negatives and false positives. We evaluated the AUC statistic to determine model accuracy and fit to the testing data (*Fielding & Bell, 1997*). The AUC statistic is a measure of the model's predictive accuracy, producing an index value from 0.5 to 1, with values close to 0.5 indicating poor discrimination and a value of 1 indicating perfect predictions (*Elith et al., 2006*). We assessed variables using response curves, variable contributions, and jackknife tests. We used percent contribution and permutation importance to determine importance of input variables in the final model (e.g., *Halvorsen 2013*). Percent contribution can be more informative with uncorrelated variables (*Halvorsen, 2013*), while permutation importance provides better variable assessment when models and variables are correlated (*Searcy & Shaffer (2016)*.

Because over-parameterized models tend to underestimate habitat availability when transferred to a new geography or time period, we used selection methods suggested by *Warren & Seifert (2011)*. Maxent provides the option of reducing overfitting with a regularization multiplier that can be altered by the user to apply a penalty for each term included in the model ($\beta$ regularization parameter) to prevent overcomplexity or overfitting (*Merow, Smith & Silander, 2013*; *Morales, Fernández & Baca-González, 2017*). A higher regularization multiplier will reduce the number of covariates in the model, becoming more lenient with an increased sample size (*Merow, Smith & Silander, 2013*). We did not include model replicates, an option in the interface, to output the required data (lambda file) and set output to logistic. We altered the Regularization Multiplier from 0.5 to 4 for each 0.5 increment (e.g., *Radosavljevic & Anderson 2014*).

We ranked candidate models using AIC corrected for small sample sizes (AIC$_c$; *Burnham & Anderson 2002*). We considered the model with the lowest AIC$_c$ value to be our top model with those with $\Delta$AIC$_c$<2 to be competitive models. For our top model, we generated predicted-to-expected (P/E) ratio curves for our model using only the testing data to evaluate its predictive performance, which was based on the shape of the curves, a continuous Boyce index (*Boyce et al., 2002*), and Spearman rank statistics. We used the predicted-to-expected curve to inform our suitability thresholds following *Hirzel et al. (2006)*. We defined unsuitable in areas where the model performed equal to or poorer than random chance (P/E $\leq$ 1) with the lower 95% confidence interval of the P/E curve overlapping 0. For predicted suitable and highly suitable locations, we divided P/E and their respective 95% confidence values greater than 1, categorizing the lower half of data as suitable and the upper portion as predicted highly suitable.

## RESULTS

### Locations

We compiled 10,229 Humboldt marten locations collected during 1996–2020 (542 baited station, 263 detection dog team, 831 VHF telemetry, 8,537 GPS telemetry, 15 roadkill, and

**Table 2** **We show the percent contribution and permutation importance from our top Maxent model.** We ordered variables by their percent contribution and report the optimized spatial scale (focal radius in meters), the univariate response type, and whether the univariate dependent plots were generally positively or negatively correlated with Humboldt marten (*Martes caurina humboldtensis*) locations.

| Variable | Scale | Response | Univariate relationship | Percent contribution | Permutation importance |
|---|---|---|---|---|---|
| Salal | 1170 | Quadratic | + | 23.3 | 15.5 |
| Percent pine | 1170 | Product | + | 22.5 | 30.3 |
| Precipitation_30-year average | 1170 | Product | + | 21.6 | 25.3 |
| Canopy cover | 1170 | Quadratic | + | 18.7 | 20.2 |
| Mast | 1170 | Product | + | 5.4 | 1.3 |
| August temperature_30-year average | 1170 | Linear | – | 4.7 | 2.3 |
| Percent slope | 1170 | Quadratic | – | 2.7 | 4.4 |
| Old growth structural index | 50 | Linear | – | 1.2 | 0.7 |

41 others). Our GPS data represented locations taken every 2.5–5 min on 7 individuals within the Central Coast (*Linnell et al., 2018*), and we did not display those clustered data. After we spatially-thinned locations, 384 locations remained and were spread among Extant Population Areas: Central Coastal Oregon (*n* = 77 locations), Southern Coastal Oregon (*n* = 77 locations), California-Oregon Border (*n* = 33 locations), and Northern Coastal California (*n* = 192 locations) (Fig. 1). There were 5 locations that did not occur within boundaries of any EPA (*USFWS, 2019*). Location types included den or rest structure locations (18%), genetically verified scats or telemetry locations (32%), and baited camera or track plate locations (50%).

Thinned locations had similar medians and data distributions to the full location dataset, except for mast and precipitation where the medians were slightly lower for the thinned locations (Fig. 2). Non-detection locations had similar medians and data distributions to random locations, with the most notable difference between medians for salal (Table 1, Fig. 2). Differences between non-detection and random locations were likely due to clustered sampling efforts (Fig. 1B).

## Distribution modeling

Our final model included 8 variables after excluding correlated variables (Tables S2, S3). Variables in our model were optimized at the home range spatial scale (1,170 m) except OGSI (50 m), but differences between scales were modest (Figs. S1–S3). Our top model had a Regularization Multiplier of 1.5. Predictor variables, in order of percent contribution, included a positive relationship with salal (23.3%), percent pine (22.5%), average annual precipitation (21.6%), canopy cover (18.7%), and mast (5.4%) followed by a negative relationship with average maximum August temperature (4.7%), percent slope (2.7%), and OGSI (1.1%, Table 2). Permutation importance was similar with the same top four variables highly contributing, but with a slightly modified order of percent pine (30.3%), average annual precipitation (25.3%), canopy cover (20.2%), and salal (15.5%; Table 2). The OGSI variable contributed least for both metrics.

We interpreted Maxent's univariate response curves and provide the marginal plots as a supplemental figure (Fig. S4). Marten locations were correlated with both low and high

amounts of canopy cover and percent slope (quadratic response, Fig. 3). Moderate amounts of canopy cover (e.g., 5–50%) appeared to be negatively correlated with marten locations. Predicted marten distribution was positively correlated with salal with some likelihood of a threshold at high values (Fig. 3), percent pine (Fig. 3), average annual precipitation (Fig. 3), and mast (Fig. 3). There was a negative correlation between marten locations and August temperature (Fig. 3) and a slightly negative to neutral relationship between marten locations and OGSI (Fig. 3).

The predicted versus expected curve of our final model delineated unsuitable areas as <14%, suitable areas as 15–30%, and predicted highly suitable at >30% predicted probability (Fig. 4, Data S3) with an AUC value on the test data at 92%. The model depicted southern Oregon and northern California as having the largest spatial extent for predicted marten distribution, including areas south of the current known distribution (Fig. 5, Data S3).

## DISCUSSION

We developed a range-wide species distribution model for the Humboldt marten based on extensive survey effort and incorporation of contemporary vegetation and climatic conditions. Our model is complementary, but not similar, to other Humboldt marten distribution models (e.g., *Slauson et al., 2019*), which could lead to confusion when attempting to understand Humboldt habitat associations. Instead of interpreting differences between models as a conflict, we posit this as evidence of the conservation challenge described by *Caughley (1994)* and representative of the difficulty in establishing patterns of causality from observational studies. Nonetheless, our model predicted areas where Humboldt martens are known to occur and identified areas of potential occurrence outside of known population extents, which can be placed within an ecological theory framework for managers. As with all models, there are limitations associated with our predictions, and a clear assessment of these constraints is critical for model results to be accurately used to inform management decisions (*Sofaer et al., 2019*).

The role of biotic interactions in shaping the distribution of species has been reported (e.g., *Forchhammer et al., 2005*; *Guisan & Thuiller, 2005*), yet evidence of the importance of biotic variables alongside abiotic variables for predicting distributions at larger spatial scales has been largely lacking (e.g., *Wisz et al., 2013*). High amounts of shrub cover appears to be the most prevalent component of Humboldt marten locations in both California (*Slauson & Zielinski 2009, Slauson, Zielinski & Hayes, 2007*) and Oregon (*Moriarty et al., 2019*) and accordingly, both salal and mast (including mast-producing shrubs) had a strong contribution to our model. Although associations with shrub cover or mast are generally uncharacteristic of martens, European pine martens may occur in areas of dense shrubs (*Lombardini et al., 2015*) and American marten population numbers in New York appear correlated with mast in hardwood forests (*Jensen et al., 2012*). Our finding that Humboldt marten distribution was strongly correlated with canopy cover is consistent with previous marten research (*Bissonette et al., 1997*, *Hargis et al., 1999*), although our response was quadratic, suggesting marten locations were associated with both low and

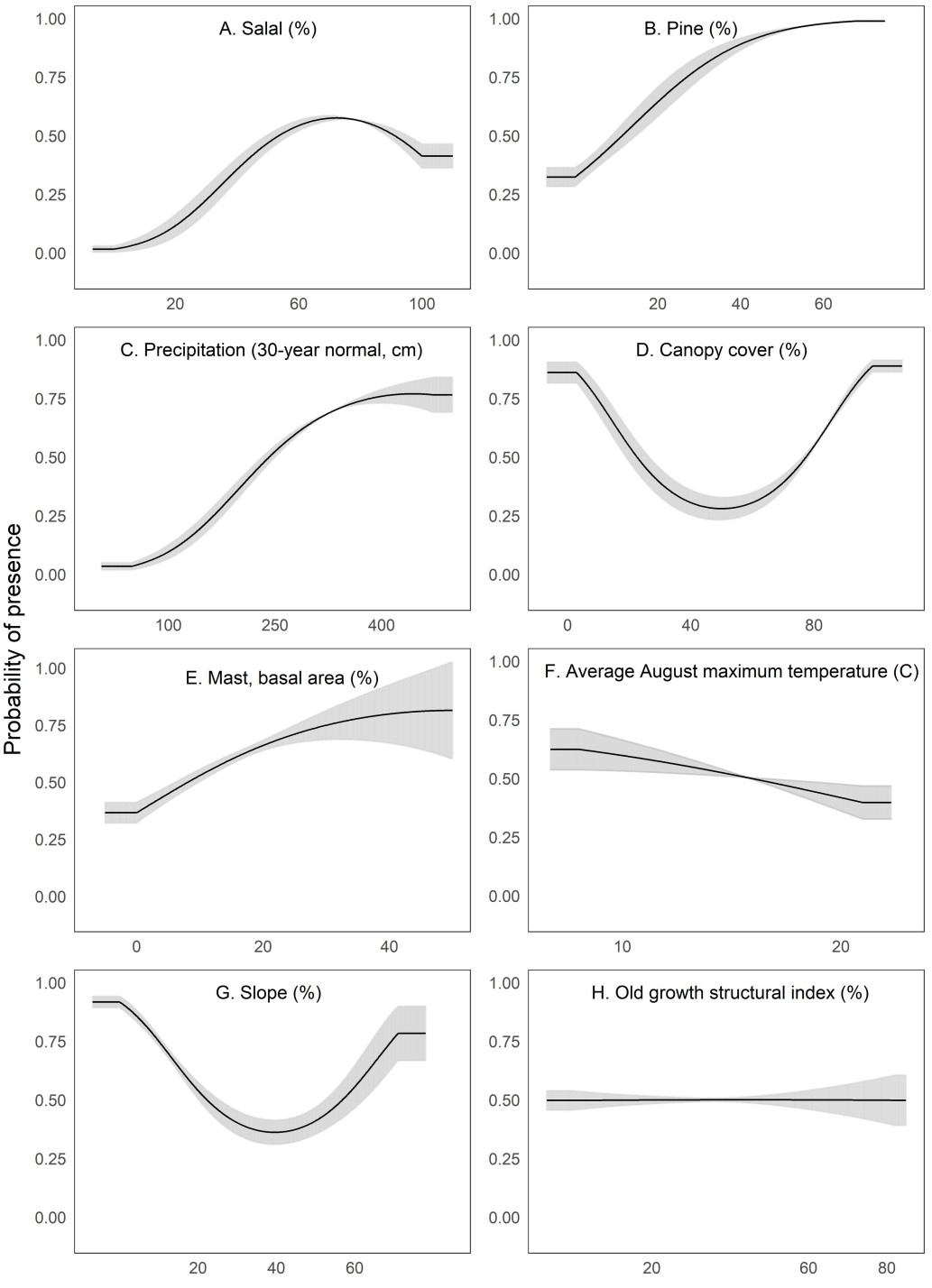

**Figure 3** **We depict predicted relationships between Humboldt marten locations and each of the variables within our final model (A-H).** Here, each curve is the predicted probability of presence with no conflicting influence of potentially correlated variables. Humboldt marten locations were correlated with both low and high amounts of canopy cover and percent slope (quadratic response). Predicted distribution was positively correlated with predicted salal (*Gaultheria shallon*) distribution, percentage of pine, precipitation, and mast. We observed a negative correlation between marten locations and August temperature. We observed a slight negative relationship between marten locations and the old growth structural index. Our figure order matches the percent contribution values reported in Table 2. The curves reveal the mean response (black) and standard deviation (gray) for 10 replicate Maxent runs.

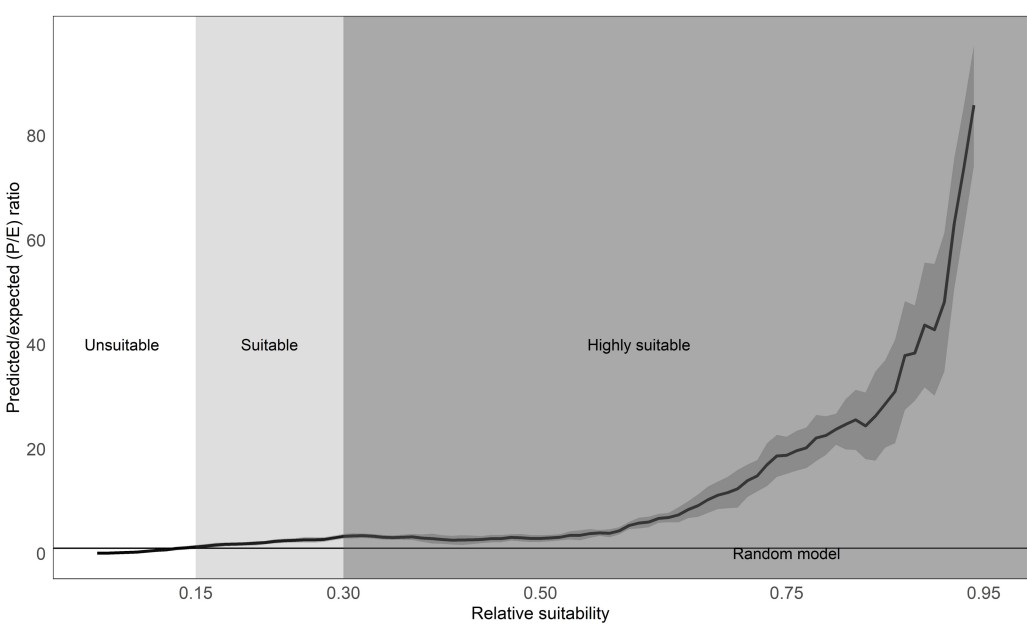

**Figure 4 Our predicted suitable transitions for Humboldt marten (*Martes caurina humboldtensis*) range.** We present mean predicted vs. expected curve (solid black line) from our model replicates, showing 95-percent confidence intervals (gray-shaded vertical bars). The P/E = 1 threshold is where the curve crosses the random chance line (horizontal orange line), and the blue dashed vertical lines are the 95-percent confidence intervals. We used the predicted-to-expected curve to inform our suitability thresholds following *Hirzel et al. (2006)*, including predicted unsuitable (P/E and confidence intervals 0–1), marginal (P/E > 1 but overlapping confidence intervals), and suitable (P/E and confidence intervals > 1; map depicted in Fig. 5).

high levels of canopy cover. Marten populations are typically associated only with relatively dense and increasing canopy cover (*Shirk, Raphael & Cushman, 2014*) and we posit that a quadratic response to canopy cover by Humboldt martens may be a function of shrub cover. Although additional information is needed to describe fine-scale vegetation associations, forest conditions with a dense understory layer of shrub and mast-producing species represent achievable targets that can guide management or restoration.

Biotic variables influencing predicted Humboldt marten distribution in our model were consistent with previous literature with some exceptions, most notably forest age and OGSI. Within our model, the predicted relationship between Humboldt marten distribution and higher OGSI values was not only weak but often negative (Supplemental Item S1). The OGSI variable may, in fact, represent an interpretive mismatch with shrub cover—some areas where Humboldt martens occur (e.g., mature Douglas fir forest; *Slauson, Zielinski & Hayes, 2007*) are characterized by both older forest conditions (i.e., high OGSI values) and substantial shrub cover, while other areas (e.g., serpentine or coastal pine forests; *Eriksson et al., 2019*; *Moriarty et al., 2019*) are characterized by substantial shrub cover, but not older forest conditions (i.e., low OGSI values). As an example of this mismatch, much of the putative distribution of Humboldt martens in coastal Oregon and California is dominated by mature western hemlock forests with high OGSI values, yet Humboldt
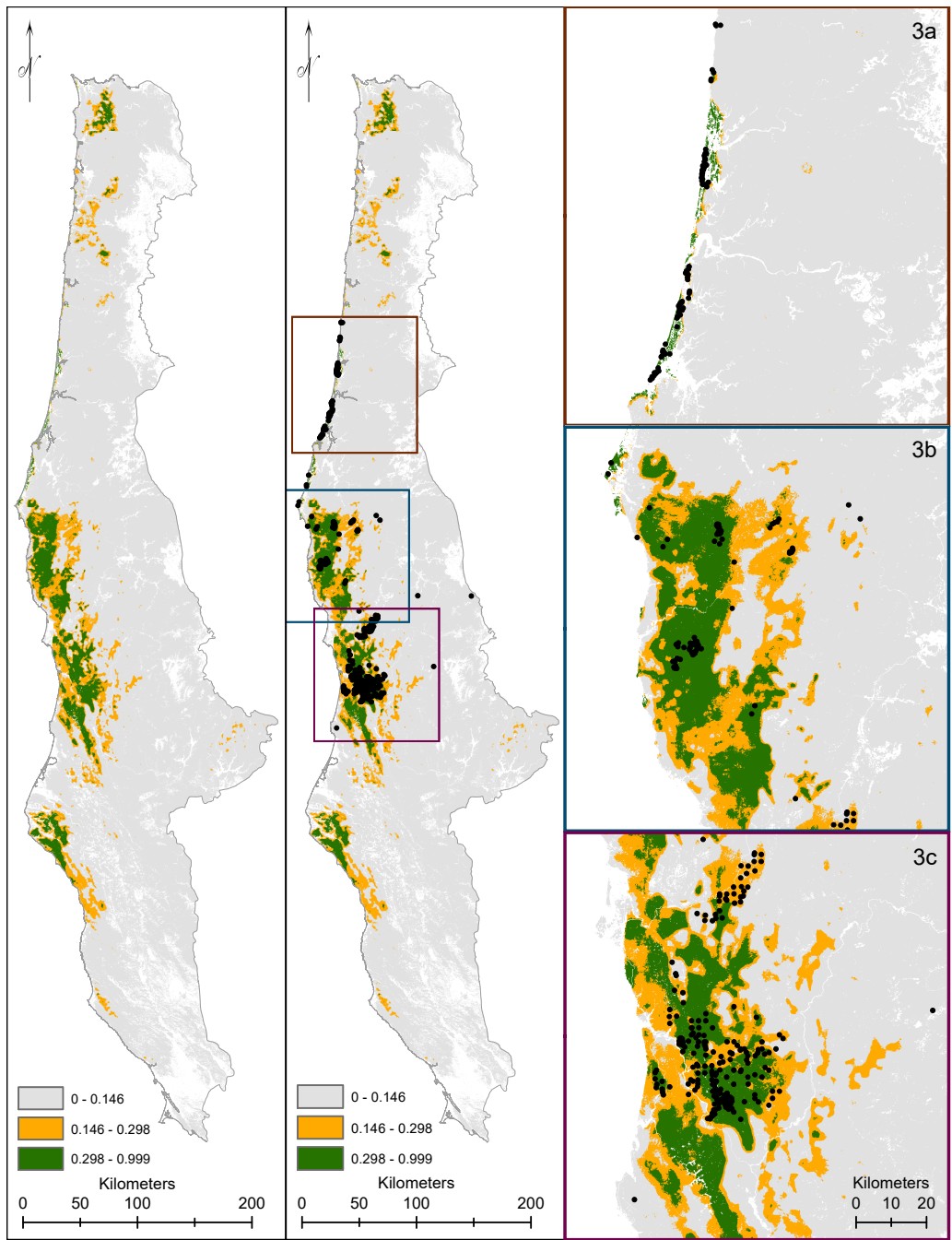

**Figure 5** **We display our modeled predicted range for Humboldt marten (*Martes caurina humboldtensis*).** For predicted range, we followed *Hirzel et al. (2006)* with predicted versus expected ratios transitioning between predicted highly suitable (green), suitable (orange), and marginal or not predicted suitable (gray). Marten location information was displayed (black dots). We zoomed to population extents to provide increased visual resolution within the Central Oregon Coast (3A), South coast (3B), and northern California (3C).

martens are not strongly associated with such areas (*Moriarty et al., 2019*), possibly because hemlocks are a shade-tolerant species that prohibit understory growth such as shrubs (*Kerns & Ohmann, 2004*). When examining our marten locations in a model only with the components of OGSI, downed wood was the most influential variable (Supplemental Item S1). We suspect the differences between our model and the *Slauson et al. (2019)* model resulted from non-stationary vegetation associations that were only revealed by increased survey effort across a broader geographic scope. While the *Slauson et al. (2019)* model relied on a modest number of Humboldt marten detections from 1996–2010 with poor coverage outside of northern California (*USFWS, 2019*), our model included a relatively large number of detections that occurred across a longer period of time (1996–2020), over a broader geographic scope in both California and Oregon (*Barry, 2018*; *Gamblin, 2019*; *Linnell et al., 2018*; *Moriarty et al., 2019*).

Range limit theorems have long postulated the importance of elevation, altitude, and weather in determining the limits of species distributions (e.g., *Darwin, 1859*). Precipitation was one of the top 3 predictive variables in all model simulations and abiotic factors such as increased precipitation, proximity to the coast, and cool temperatures likely influence vegetation type and composition. If these variables are causally linked to marten occurrence, a plausible mechanism is that cooler wetter conditions result in dense vegetation growth (e.g., shrubs). In areas with relatively low canopy cover but dense shrubs, shrub cover may be functionally similar to canopy cover by offering increased protection from predators (*Hawley & Newby, 1957*). High shrub cover also likely results in an increased availability of berries and mast. Given that martens consume prey items (e.g., birds, rodents) that feed on berries and mast, while also consuming berries themselves, shrubs may both indirectly and directly subsidize marten diets. If increased shrub cover decreases predation risk while simultaneously providing abundant food resources, such areas may provide exceptional, if uncharacteristic, marten habitat (*Eriksson et al., 2019*). If this is a potential mechanism, an example includes the abundance of huckleberries that have been attributed to increased reproduction and population growth for grizzly bears (*Ursus arctos*) over a 32-year investigation (*McLellan, 2015*).

Species' distributions may also be strongly influenced by less-apparent factors such as interspecific interactions with predators or competitors (*Siren, 2020*). As an example, spotted owls (*Strix occidentalis*) closely align with old-growth forest conditions which have been characterized with relatively high accuracy (*Davis et al., 2016*), yet spotted owl population viability is dramatically decreased with presence of barred owls (*S. varia*) due to interspecific competition and predation (*Wiens, Anthony & Forsman, 2014*; *Diller et al., 2016*; *Dugger et al., 2016*). Although few examples exist for carnivores, a recent evaluation suggests that while lynx (*Lynx lynx*) distributions are closely-tied to deep snow, the influence of reducing bobcat (*L. rufus*) competition was stronger than the influence of snow itself (*Siren, 2020*). A directed research effort that integrates the influence of vegetative and climatic associations with other factors such as prey availability, predation pressure, and competition would provide better insight on the drivers of Humboldt marten occurrence and a more holistic determination of marten distribution.

Our results predict some of the components that comprise suitable marten habitat but describing optimal habitat would be best informed by measures of survival and fecundity. Future endeavors could develop site-specific models, ideally using telemetry data that are biologically linked with fitness (e.g., long-lived adult female rest and den structures) to address predicted habitat. We lack enough information regarding where Humboldt martens resided historically to compare with our contemporary distribution (*Loehle, 2020*), and we are generally ignorant of population densities, causal associations of population declines, and population limitations. Such an understanding is essential to describe expectations of future range (*Brown, Stevens & Kaufman, 1996*). Finally, the lack of consistency among Humboldt marten studies is suggestive of imperfect knowledge of what components constitute Humboldt marten habitat. To avoid differing views for rare species conservation (e.g., *Gutiérrez, 2020*; *Jones et al., 2020*), amassing information collaboratively with a goal of prospective meta-analyses and study-level replication will be essential (*Facka & Moriarty, 2017*; *Nichols, Kendall & Boomer, 2019*).

## CONCLUSIONS

Based on our modeling and an evaluation of available evidence, we conclude that the most consistent range-wide characteristic with Humboldt marten distributions are forest associations with extensive dense shrub cover or complex understory vegetation, which may be indicative of increased food availability or predation escape cover. An understanding of the strength of these interactions and factors that limit populations is needed to make informed conservation decisions. An adaptive management framework with integrated research components may allow for near-term conservation decision making.

## ACKNOWLEDGEMENTS

The desire and decision to request an updated model incorporating newer presence and habitat information was from the Oregon Humboldt Marten stakeholder group, which is facilitated by the US Fish and Wildlife Service in Oregon. Non-invasive marten surveys were conducted by Pacific Northwest and Southwest Research Stations, Oregon State University, Humboldt State University, Green Diamond Resource Company, NCASI, the Siuslaw, Rogue-Siskiyou, and Six Rivers National Forests, Hancock Forest Management, Weyerhaeuser, Oregon Department of Forestry, and the Confederated Tribes of Siletz Indians of Oregon. Detection dog surveys were completed by Rogue Detection Dog Teams and the former group within Conservation Canines, University of Washington. Considerable aid with field logistics, vehicles, housing, and equipment was provided by the US Fish and Wildlife Service, Salem District BLM, USFS Rogue River-Siskiyou and Siuslaw National Forests, Weyerhaeuser, Hancock Forest Management, and USFS Region 6 Regional Office. We obtained private land access or surveys were completed by trained staff within the ownership for all randomly selected survey points—thanks to Weyerhaeuser, Hancock Forest Management, Starker Forests, and Roseburg Timber for access or data. Reviews by Drs. E. Forsman, D. Miller, and J. Verschuyl, A. Balestrieri, B. Hollen, our anonymous peer reviewer, and the Associate Editor G. Casazza, improved

previous versions of this manuscript. Extreme thanks to all field crew leaders (S. Smythe, M. Linnell, B. Peterson, G. W. Watts, J. Bakke, C. Shafer, K. Kooi, and M. Penk) and team members (E. Anderson, D. Baumsteiger, A. Benn, J. Buskirk, B. Carniello, M. Cokeley, S. Hart, P. Iacano, A. Kornak, T. McFadden, E. Morrison, A. Palmer, T. Peltier, N. Palazzotto, S. Roon, S. Riutzel, C. Scott, K. Smith, R. Smith, T. Stinson, M. Williams, B. Woodruff, and K. Wright). Any use of trade, firm, or product names is for descriptive purposes only and does not imply endorsement by the US Government.

### Funding

Marten survey efforts were funded by the Oregon State University Fish and Wildlife Habitat in Managed Forests Research Program, NCASI, Coos Bay-Bureau of Land Management (BLM), the USDA Forest Service (USFS) Siuslaw and Rogue-Siskiyou National Forests and Six Rivers National Forest, the Oregon Forestry Industry Council, U.S. Fish and Wildlife Service (Arcata office), and Humboldt State University Sponsored Programs Foundation. NCASI, USFS Pacific Northwest and Southwest Research Stations funded genetic confirmation of scats, remote camera data processing and management, and provided external support. Telemetry data were funded by the Siuslaw National Forest, Green Diamond Resource Company, and U.S. Fish and Wildlife Service. The funders had no role in study design, data collection and analysis, decision to publish, or preparation of the manuscript.

### Grant Disclosures

The following grant information was disclosed by the authors:
Oregon State University Fish and Wildlife Habitat in Managed Forests Research Program. NCASI.
Coos Bay-Bureau of Land Management (BLM).
The USDA Forest Service (USFS) Siuslaw and Rogue-Siskiyou National Forests and Six Rivers National Forest.
The Oregon Forestry Industry Council, U.S. Fish and Wildlife Service (Arcata office), and Humboldt State University Sponsored Programs Foundation.
NCASI, USFS Pacific Northwest and Southwest Research Stations.
The Siuslaw National Forest, Green Diamond Resource Company, and U.S. Fish and Wildlife Service.

### Competing Interests

Jennifer Hartman is employed by Rogue Detection Teams. Keith Hamm & Desiree Early are employed by Green Diamond Resource Company. Authors in academic or research positions (e.g., Moriarty, Linnell, Delheimer, Levi, Gunther, Gamblin) may gain notoriety or career advancements with peer-reviewed publications. Authors associated with land management agencies may need to execute or suggest restoration activities in areas designated by the model (e.g., Barry, Davis) or conduct additional surveys (e.g., Hartman, Ellison, Moriarty).
## Author Contributions

- Katie M. Moriarty and Mark Linnell conceived and designed the experiments, performed the experiments, analyzed the data, prepared figures and/or tables, authored or reviewed drafts of the paper, and approved the final draft.
- Joel Thompson analyzed the data, prepared figures and/or tables, authored or reviewed drafts of the paper, and approved the final draft.
- Matthew Delheimer, Brent R. Barry, Taal Levi, Keith Hamm, Desiree Early, Holly Gamblin, Micaela Szykman Gunther and Jordan Ellison conceived and designed the experiments, performed the experiments, authored or reviewed drafts of the paper, and approved the final draft.
- Janet S. Prevéy analyzed the data, authored or reviewed drafts of the paper, created predicted models for salal and vaccinium, and approved the final draft.
- Jennifer Hartman performed the experiments, authored or reviewed drafts of the paper, and approved the final draft.
- Raymond Davis analyzed the data, authored or reviewed drafts of the paper, and approved the final draft.

## Animal Ethics

The following information was supplied relating to ethical approvals (i.e., approving body and any reference numbers):

Our protocols were reviewed by the USDA Forest Service Research and Development Institutional for Use and Care Committee. We received approval for marten captures and treatment in the Central Coast (USDA FS R&D 2015-002). We received a waiver for camera trapping by the same committee (USDA FS R&D 2017-005). In California, Gamblin's 2018-2019 non-invasive field surveys were conducted in accordance to Humboldt State University's IACUC Permit number 16/17.W.05-A.

For the data for which the authors were responsible, our protocols were reviewed and approved by the USDA Forest Service Research and Development Institutional Care and Use Committee (permits 2015-002, 2017-005) or Humboldt State University Institutional Care and Use Committee (permit 16/17.W.05-A).

## Field Study Permissions

The following information was supplied relating to field study approvals (i.e., approving body and any reference numbers):

We compile several projects that had permission to collect non-invasive hair samples under the USDA Forest Service Research and Development Institutional Care and Use Committee, Humboldt State University's Institutional Care and Use Committee, the Oregon or California Department of Fish and Wildlife.

We obtained Scientific Take Permits for hair snares and samples collected through the Oregon Department of Fish and Wildlife (ODFW 119-15, 128-16, 033-16, 109-19, 107-20). Older verified survey data were provided by the US Fish and Wildlife Service with no additional information.

## Data Availability

Data are available in the Supplemental Files and at Dryad: https://datadryad.org/stash/dataset/doi:10.5061/dryad.qnk98sfgt.

## Supplemental Information

Supplemental information for this article can be found online at http://dx.doi.org/10.7717/peerj.11670#supplemental-information.

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
