# Peer review of "Predicted distribution of a rare and understudied forest carnivore: Humboldt marten (Martes caurina humboldtensis)"

_PeerJ, doi:10.7717/peerj.11670_

## Round 0.1 · original submission · Major Revisions

· Academic Editor

Major Revisions

Dear Dr. Moriarty,

Overall the reviewers find you made a high research effort using an impressive amount of data. Nevertheless, they found the manuscript lacked clarity in several parts, making sometimes the text confusing and difficult to appreciate the impact of your research. For this reason, they suggest reorganizing, revising, and shortening the text.

Please, respond point-to-point to the comments of reviewers to speed up the process of revision.

Once again, thank you for submitting your manuscript to PeerJ and we look forward to receiving your revision.

Sincerely,
Gabriele Casazza

·

Basic reporting

The authors’ research effort was high but, despite being suitable for the aims and scope of PeerJ, I regret to say that in this form the ms. cannot be accepted for publication, needing to be deeply re-thought, revised and shortened for the sake of conciseness and clarity.
About shortening, in the attached PDF I have suggested the paragraphs that should be cut, because useless, repetitions or too speculative. This said, the authors need to put order in the methods and discussion sections, trying to be brief, which will result in being also straightforward and clearer. Below (General Comments) I suggest several changes which, in my intention, may help to revise the ms.

Experimental design

Although often difficult to follow, on the whole methods and analyses seem appropriate.
Models were run at 4 different scales (buffer radius), to test at which scale each variable was most strongly related to the species presence; anyway, in the results section, details about the predictive performance of the 4 best models are lacking.

Recent literature about HSMs for the pine marten should be included for comparing patterns

Validity of the findings

Lack of clarity throughout the ms makes it difficult to appreciate the soundness and impact of the research, but i believe that, when deeply revised, the ms will provide useful and interesting results

Additional comments

L46-49: this sentence disagrees with L500-503
L55-56: please see my comment at L124-25 and 536-60
L56-58: do you mean: “martens were recorded also in areas filed as unsuitable by the model”?
L50-51: is salal a shrub or a tree?
L68-94: this “incipit” let the reader think that your study will in some way overcome the described obstacles/bias, but this hope will not be satisfied. I suggest focusing on the need for detailed info about the distribution and habitat preference of elusive and low-density species such as H martens for conservation management.
L70-72: need rephrasing.
L95-97: need rephrasing.
L105-112: I suggest referring first to broad-scale attributes (large home range) and then to fine-scale ones.
L124-5: I would rather say that recent studies have highlighted that martens are more generalist than we believed. Also the pine marten had been associated to old and well-structured forests, but currently it has colonised small wood patches in lowland areas of SW Europe, where shrub cover sometimes becomes a major variable affecting its presence
L155: needs rephrasing.
L163: state clearly that you sampled all the 4 populations (if you did)
L191-2: need rephrasing.
L193: This 500 m cell has not been introduced before; you should state that you used a grid (UTM?)
L194: state here, rather than at L202, that you used only one location per cell.
L205: need rephrasing.
L237-8: repetition of L225-8
L239-40: lacking details or references, it is not clear in which way you obtained forest age.
L241 Canopy percent cover
L236-40: be more direct, for the sake of clarity: “We incorporated N biotic variable: 1,2,3,4. The first was assessed as …
L248: are these the min-max values within which the 12 diameter thresholds fall?
L252: please use “…” for variable names (e.g. “percent pine”) throughout the main text
L255-58: more simply: “because martens have been shown to occur in sparse pine woods growing on serpentine soils in the OCC” (right?)
L262: “overlap … areas”: unclear
L269: “which was the sum of the probabilities of both species occurrence” (?)
L290: basal area = cover?
L295: as for L236
L303-9: you may explain this var more clearly and briefly, e.g. “Using 4-km resolution available records from May through September 1996–2017 (NASA/NOAA Geostationary Environmental Satellite Imager measurements), “coastal low cloudiness” was satellite derived as described by Clemesha et al. (2016) and quantified as the yearly mean (± SD) percent of time that low clouds were present relative to the number of valid half-hourly, daily observations”.
L327: For clarity, before describing home ranges, state that the two largest radius were assessed from telemetry data. If these data have been already published, you may be briefer (L327-37).
L354-360: did you use Pearson or VIF? For the latter, usually a VIF>3 threshold is used to select variables
L399: “when… period” needs rewording.
L432-36: locations don’t seem evenly distributed (6, 37, 3, 54%)
L441-3: Both the medians and data distributions of all variables did not differ between the thinned sub-sample and the whole data set, except for…. Similarly, surveyed locations with no marten detections and random locations differed only for “salal”…
L450-1: based on what data/comparisons do you state this?
L452: I would say “suggesting that most variables… at the hr scale.”
L464-5. Comment to be moved to the discussion.
L472: line difficult to read and understand; do you mean: The predicted versus expected curve of our final model…?
L475: do you mean “the largest areas suitable to martens are in S Oregon and N California?”
Discussion: I suggest to start the discussion commenting biotic variables. Then, go on with lines 497-503 and add comments on abiotic variables
L484-5: but these abiotic variables may address conservation actions, e.g. providing protection areas including suitable (cool and wet) habitats
L:504-510: very confused and hard to understand
L512-13: “…related to increase canopy cover” but wasn’t the response of this variable quadratic (L506)?
L536-60: you should refer to the similar results that have been found for the pine marten in Mediterranean Europe (Spain, France, Italy), which suggest that martens are less forest-specialist than previously believed.
L597: here you tell us again about temp and rain, which had been commented at the start of the discussion!
L609-11: what do you mean?
L634-35: = L 479-81
Figure 3: I suggest showing response curves in order of variable importance for the model (i.e. start with salal and end with OGSi)

Reviewer 2 ·

Basic reporting

This is an overall well-structured and clear manuscript, however, it is sometimes confusing and long in certain sections.

Experimental design

It is impressive the amount of data the authors are using to conduct the modeling presented here, which mostly relies on the collection of presence records and the derivation of environmental predictors used in modeling. However, the objective of the work or its main question is confusing: in the abstract, they refer to the historical distribution of the species, and this is later in the following sections not to be the case (including the Discussion). This is confusing overall.
Parts of the methods need to be reviewed to be clear (highlighted as specific comments in the PDF), for example, the meaning of some variables and specifications in the use of Maxent. Also, the accessibility area seems not to be completely adequate and should be carefully justified from a biogoegraphical point of view (see comments on PDF).
The Discussion is too long and repetitive at times. I suggest you organize by type of environmental predictor (e.g., biotic/abiotic) and then discuss how these variables are or could be interchanchagle surrogates. How this connects to Eltonian and Grinnellian niches may be also attractive.

Validity of the findings

All areas of the manuscript need to be reviewed and used as an opportunity to frame the problem and questions and then their results and discussion.
There are probably many conclusions throughout the Discussion that are probably lost because of the extensive length of this section. This should be reviewed and organized better. Overall, this sounds more like a really nice and modern natural history exercise improved by finely tuned ecological niche modeling methods. Also, try to clarify whether your work is about (or when is this the case to avoid confusion) a historic reconstruction of the distribution of martens or more on the ecology and natural history perse. It is obviously both or all of them, but better framing should help to improve transmitting the key findings.

Additional comments

Please, use the comments on the previous sections as a reference to general comments and also use the attached PDF to look at specific comments and areas where I believe your manuscript could see improvement.

Annotated reviews are not available for download in order to protect the identity of reviewers who chose to remain anonymous.

---

## Round 0.2 · Minor Revisions

· Academic Editor

Minor Revisions

Dear Dr. Moriarty,

The reviewer thinks your manuscript has been greatly improved. He only suggests some minor corrections aimed to further improve the paper. Reviewer comments and suggestions are in the attached pdf file. Please, respond point-to-point to the comments of reviewers to speed up the process of revision.

Once again, thank you for submitting your manuscript to PeerJ.

Sincerely,
Gabriele Casazza

·

Basic reporting

no comment

Experimental design

no comment

Validity of the findings

no comment

Additional comments

I am glad to say that he ms has been greatly improved. There are some minor corrections that may further improve the paper (see the attached file)

---

## Round 0.3 · accepted · Accept

· Academic Editor

Accept

Dear Dr. Moriarty,

I am very pleased to inform you that your paper "Predicted distribution of a rare and understudied forest carnivore: Humboldt martens (Martes caurina humboldtensis)" is accepted for publication in the PeerJ. Congratulations!

Thank you for submitting your work to PeerJ.

Sincerely,
Gabriele Casazza